# Stress Response Is the Main Trigger of Sporadic Amyloidoses

**DOI:** 10.3390/ijms22084092

**Published:** 2021-04-15

**Authors:** Alexey P. Galkin, Evgeniy I. Sysoev

**Affiliations:** 1St. Petersburg Branch, Vavilov Institute of General Genetics, 199034 St. Petersburg, Russia; 2Department of Genetics and Biotechnology, St. Petersburg State University, 199034 St. Petersburg, Russia; st063251@student.spbu.ru

**Keywords:** amyloid, sporadic amyloidoses, stress response, trigger of amyloidosis, protein overproduction, modifications

## Abstract

Amyloidoses are a group of diseases associated with the formation of pathological protein fibrils with cross-β structures. Approximately 5–10% of the cases of these diseases are determined by amyloidogenic mutations, as well as by transmission of infectious amyloids (prions) between organisms. The most common group of so-called sporadic amyloidoses is associated with abnormal aggregation of wild-type proteins. Some sporadic amyloidoses are known to be induced only against the background of certain pathologies, but in some cases the cause of amyloidosis is unclear. It is assumed that these diseases often occur by accident. Here we present facts and hypotheses about the association of sporadic amyloidoses with vascular pathologies, trauma, oxidative stress, cancer, metabolic diseases, chronic infections and COVID-19. Generalization of current data shows that all sporadic amyloidoses can be regarded as a secondary event occurring against the background of diseases provoking a cellular stress response. Various factors causing the stress response provoke protein overproduction, a local increase in the concentration or modifications, which contributes to amyloidogenesis. Progress in the treatment of vascular, metabolic and infectious diseases, as well as cancers, should lead to a significant reduction in the risk of sporadic amyloidoses.

## 1. Introduction

Amyloids are fibrillar proteins possessing a cross-β conformation in vivo. Some proteins function normally in the amyloid form in prokaryotes and eukaryotes, while the formation of amyloid fibrils of other proteins is associated with the development of diseases called amyloidoses. Pathological amyloid fibrils are formed as a result of protein misfolding, which can be provoked by various factors. Almost all amyloidoses are incurable, and millions of people die from these diseases every year. With an increase in life expectancy, the number of people dying from common amyloidoses is increasing exponentially. All pathological amyloidoses can be divided into three main groups, depending on their origin: infectious, genetically determined and sporadic. Our review will consider the causes of sporadic amyloidoses, but first we will briefly characterize all three groups of these diseases.

Infectious amyloidoses, or prion diseases, are associated with the body-to-body transmission of fibrillar prion protein (PrP) particles. The amyloid isoform of this protein is called PrP^Sc^ (Sc stands for scrapie, the first reported prion disease in sheep) [1]. Prion fibrils are transmitted between mammalian species and cause irreversible neurodegeneration. Interspecies transmission of PrP^Sc^ particles has been shown when intracerebral inoculation, intravenous injection, or oral administration of brain homogenate from one species causes infection in animals of another species [2,3,4]. Prion fibrils demonstrate remarkable resistance to various factors [5]. They also are uncleaved by digestive enzymes in the gastrointestinal tract [5]. PrP^Sc^ aggregates penetrate lymphoid organs to be accumulated in the follicular dendritic cells, where foreign infectious particles induce the conversion of the cellular PrP^C^ (C stands for cellular) into the aberrant PrP^Sc^ isoform. Next, the resulting PrP^Sc^ oligomers are transported from the follicular dendritic cells to the cells of the peripheral and then central nervous system [5]. PrP^Sc^ particles convert neuronal PrP^C^ in the brain into an abnormal shape and cause neurodegeneration [5,6]. The prion disease called kuru was common among natives of Papua New Guinea due to ritual cannibalism [2]. Prion diseases also include the mad cow disease, which caused significant economic damage to agriculture in Europe at the end of the last century [7]. Cases of death of people who ate the meat of sick cows have been registered [8]. Certain infectious properties have been shown for some other amyloid proteins, such as α-synuclein and tau [9,10,11,12], but the natural foodborne transmission is indicated only for PrP^Sc^.

The group of amyloidoses associated with mutations that promote protein aggregation is extensive. Mutations that cause a predisposition to the development of amyloidoses can be inherited, as well as occur de novo in somatic cells. For example, a number of neurodegenerative amyloidoses, associated with mutations in the gene encoding the prion protein, have been characterized [13]. Mutations in several genes have been shown to contribute to the accumulation and aggregation of amyloid peptide beta (Aβ), which causes neurotoxicity in Alzheimer’s disease [14]. These mutations lead to the development of early-onset familial Alzheimer’s disease [15]. Some proteins can form amyloid aggregates only against the background of certain mutations [16,17]. In particular, Huntington’s disease is caused by an expanded CAG trinucleotide repeat encoding a tract of consecutive glutamines near the amino terminus of the huntingtin protein [18,19]. The polyglutamine sequence is cleaved from the protein and forms neurotoxic intracellular aggregates. It should be noted that all infectious and genetically determined forms affect only 5–10% of people suffering from amyloid diseases.

The most common group of amyloid diseases are so-called sporadic amyloidoses (Table 1). Amyloidoses that belong to this group are associated with the aggregation of wild-type proteins. Some sporadic amyloidoses are known to be induced only against the background of certain pathologies, but in some cases the cause of amyloidosis is not so obvious. It is assumed that these diseases often occur by accident. In this review, we have summarized current data on all sporadic amyloidoses and analyzed the factors that may contribute to the formation of cytotoxic aggregates. First of all, the formation of amyloid fibrils is facilitated by an increase in production levels or local protein concentration. In addition, in some cases, amyloidogenesis occurs against the background of certain protein modifications. It is known that various pathologies cause a stress response, which is accompanied by changes in the level of production and modifications of a number of proteins. The same effects cause violations of the cell differentiation program in carcinogenesis. Generalization of current data presented in this review shows that stress response and carcinogenesis are the primary triggers for amyloidogenesis. 

## 2. Vascular Pathologies, Brain Injuries, and Oxidative Stress Trigger Amyloid Diseases

### 2.1. Sporadic Alzheimer’s Disease and Tauopathies

Brain trauma, vascular disease, and oxidative stress are risk factors for a number of widespread amyloidoses including Alzheimer’s disease (AD) [43,44,45]. Alzheimer’s disease affects approximately 17% of people aged 65–74 and 36% of people over 86 [46]. With an increase in life expectancy, the number of people suffering from this incurable disease will steadily increase. According to the World Health Organization, this disease is included in the top ten pathologies leading to fatal consequences. AD is characterized by the progressive loss of neurons and synapses in the cortex and some other regions of the brain, which leads to their gradual atrophy [47]. The etiology of AD is associated with the formation in the brain of neurotoxic intercellular inclusions of amyloid peptide beta (Aβ) and intracellular aggregates of the tau protein [48]. Aβ peptide is formed by the processing of the amyloid precursor protein (APP) [49]. The accumulation and aggregation of Aβ occurs with an increase in the level of APP production and with abnormal activation of α- and γ-secretases [49]. The incidence of AD correlates with age, but some older people have no signs of this disease. This suggests that AD is most likely not a natural age-related disease but may be caused by certain pathologies that often arise in the aging brain. It was established that stroke and vascular pathologies are risk factors for AD [50]. Postmortem analysis of the cerebral cortex of young people who died as a result of traumatic brain injuries shows diffuse Aβ plaques similar to those found in AD patients, located in the areas surrounding the lesion sites [43,51]. Moreover, oxidative stress leads to increased production and aggregation of the Aβ peptide in cell cultures [20]. At the same time, the data obtained in cell cultures cannot always be extrapolated to events that occur in the multicellular organism. Surprisingly, direct evidence that Aβ aggregation can be induced by physiological changes has been obtained in a study of the brain of Atlantic salmon grown for spawning. Hormonal changes lead to changes in morphology and at the last stage of spawning to tissue degeneration and the death of both females and males. In particular, neurodegeneration of spawning Atlantic salmon is associated with the formation of Aβ aggregates in the brain [52,53]. These data prove that the accumulation and aggregation of Aβ is not a random event but can be provoked by physiological changes. Obviously, in this case, Aβ aggregation is a consequence of hormonal changes that shift and disbalance the production of a number of proteins.

Intracellular aggregates of the microtubule-associated protein tau are another AD biomarker. Soluble isoforms of this protein participate in maintaining the stability of microtubules in axons and are abundant in the neurons of the central nervous system [54]. The hyperphosphorylated aggregates of tau are the major component of neurofibrillary tangles in clinically heterogeneous neurodegenerative disorders that are now collectively termed tauopathies [55]. Tauopathies, including but not limited to Alzheimer’s disease, progressive supranuclear palsy, corticobasal syndrome, some frontotemporal dementias, and chronic traumatic encephalopathy, are progressive neurodegenerative disorders [11]. Hyperphosphorylation and aggregation of tau causes the destabilization of microtubules, interrupting axonal transport and synaptic dysfunction [22]. The factors and mechanisms contributing to the formation of neurotoxic aggregates of this protein have been studied in sufficient detail. Experimental traumatic brain injury in rats causes a decrease in the activity of alkaline phosphatase, which dephosphorylates tau. It promotes hyperphosphorylation of the tau protein, its dissociation from microtubules, and aggregation [22]. It is assumed that chronic inflammation after brain injuries in humans leads to the same consequences [56]. Hyperphosphorylated tau accelerates stress granule formation and modulates the patterns of protein interactions of TIA1, the key stress granule component [57]. On the other hand, the formation of neuronal stress granules promotes tau aggregation [58]. All these data indicate that tau aggregation is not a sporadic event. Stressful conditions provoke the formation of cytotoxic aggregates of hyperphosphorylated tau.

### 2.2. Sporadic Parkinson’s and Prion Diseases

Another neurodegenerative disease, Parkinson’s disease (PD), is associated with the formation of amyloid aggregates of α-synuclein, and degeneration of dopaminergic neurons in the substantia nigra. The symptoms of PD usually do not develop until 70–80% of dopaminergic neurons have already been lost [59]. The sporadic form of PD typically occurs in people over the age of 60. The first symptoms are tremors, rigidity and impaired movement. Cognitive and behavioral problems occur in the advanced stages of the disease [60]. One of the diagnostic signs of this disease is the presence of cytoplasmic inclusions known as Lewy bodies in the substantia nigra [61]. Lewy bodies include several different proteins, but fibrillar aggregates of α-synuclein are the main component of these cytoplasmic inclusions [61]. Several studies have shown that α-synuclein oligomers are toxic to cells, possibly through physical disruption of cellular membranes, while Lewy bodies do not have such properties [62,63,64]. Brain injuries cause a dramatic increase in the production of α-synuclein [21]. Potential risk factors also include environmental toxins and oxidative stress [65,66]. It is shown that lipid peroxidation in chronic oxidative stress prevents fibrillation of α-synuclein and supports the formation of secondary beta sheets and toxic soluble oligomers in a dose-dependent manner [67,68]. There are many studies of the relationship between PD and various vascular pathologies, but their data are often conflicting.

The prion protein (PrP) is also overexpressed under stressful conditions. In particular, increased insulin levels and oxidative stress are reported to cause up-regulation of PrP expression [23,26]. Moreover, PrP expression is increased under hypoxic conditions in cell cultures [24,25]. It is likely that various stress factors can provoke overproduction and aggregation of PrP in the event of such non-inherited variants of prion diseases as Creutzfeldt–Jakob disease and fatal insomnia.

### 2.3. Sporadic Amyloidoses Localized Outside the CNS

The expression of another amyloidogenic peptide, known as atrial natriuretic factor (ANF), is increased in vascular pathologies. It is a peptide hormone secreted from the cardiac atria that in humans is encoded by the NPPA gene [69]. The function of ANF is to lower blood volume, reducing cardiac output and systemic blood pressure [70]. Cardiac stress not only induces secretion of stored ANF but also enhances the NPPA gene expression. Hypertension, myocardial infarction, cardiomyopathy and valve insufficiency provoke a significant increase in the ANF level [32]. Overproduction of ANF in heart disease leads to the formation of amyloid aggregates of this protein, which are localized in the cardiac atria [71]. Much less is known about the factors causing amyloidogenesis of the transthyretin (TTR) protein, which also forms pathological aggregates in the heart and other organs. Normally, TTR is a carrier for thyroid hormones and the retinol-binding protein [72]. Pathological aggregation of wild-type TTR causes senile systemic amyloidosis (SSA). Autopsy data show that the prevalence of senile cardiac amyloidosis is approximately 10% over the age of 80 years, and 50% over the age of 90 years [73]. For unknown reasons, this amyloidosis occurs 25–50 times more often in men than in women [74]. The factors causing SSA are not well understood, but it is clear that the production of TTR depends on physiological factors. In cell cultures, glucocorticoids up-regulated TTR expression, an effect suppressed by glucocorticoid receptor and mineralocorticoid receptor antagonists [33]. Moreover, induction of psychosocial stress increased TTR expression in the liver of animals subjected to acute and chronic stress conditions [33]. It is also known that age-related oxidative modifications of transthyretin modulate it’s amyloidogenicity [75].

Remarkably, vascular pathologies can trigger amyloidogenesis of the same proteins in completely different organs and tissues. To illustrate this statement, let us consider protein amyloidogenesis in preeclampsia. Preeclampsia is a multisystem pathology of pregnancy remaining a leading cause of maternal and perinatal morbidity and mortality globally [76]. Symptoms of preeclampsia include hypertension, proteinuria, edema, and maternal organ dysfunction [77,78]. According to the modern concept, the main cause of this disease is a disturbance of the angiogenesis in the placenta [79], which provokes multisystem disorders in the mother’s body and in the development of the fetus. Ischemia, hypoxia, and production of proinflammatory cytokines in preeclampsia [80] lead to endoplasmic reticulum (ER) stress [80,81] and protein misfolding [82]. The urine of patients with preeclampsia contains protein aggregates of alpha-1 antitrypsin, albumin, immunoglobulin light chains, ceruloplasmin, interferon-inducible protein 6–16 (IFI6), and Aβ peptides [76,83]. They bind the amyloid-specific dye Congo red and show a yellow-green birefringence in polarized light [83]. It is important to remember that specific in vivo staining with Congo red is a classic characteristic of amyloid proteins [84]. These protein aggregates in urine are also detected with amyloid-specific antibodies [83]. Thus, the accumulation and aggregation of the Aβ peptide occurs both in the brains of people with AD and in the bodies of pregnant women with preeclampsia. Obviously, the stress response of different cell types can lead to the accumulation and aggregation of the same proteins.

## 3. Different Types of Cancer Cause Amyloidogenesis of Certain Proteins

Any type of tumor is accompanied by the shutdown of the cellular differentiation program and massive changes in protein production. In this regard, cancer is one of the main potential risk factors for the onset of amyloidoses, since malignant transformation can increase the production of amyloidogenic proteins. Currently, several amyloidoses specific for various types of cancer have been characterized.

First of all, it is necessary to mention the immunoglobulin light and heavy chain amyloidoses, often occurring against the background of multiple myeloma. These amyloidoses are known as AL and AH, respectively [34]. Multiple myeloma and Waldenström’s macroglobulinemia cause malignant multiplication of B cells that produce monoclonal immunoglobulins. AL amyloidosis, in contrast to AH, is widespread; its estimated incidence ranges around 10 to 12 cases per million person-years [85]. Interestingly, multiple myeloma is not always associated with amyloidosis. It is estimated that approximately 15% of patients with multiple myeloma have coexisting AL amyloidosis [86]. In all likelihood, different light and heavy chains of immunoglobulins that accumulate in serum against the background of myeloma have different amyloidogenic potential.

Various types of cancer cause the overproduction of the serum amyloid A (SAA) protein [27]. This protein plays a role in the immune response to both carcinogenesis and infectious diseases [87]. SAA is synthesized as a precursor by hepatocytes in response to transcriptional stimuli from various pro-inflammatory cytokines [88]. Various diseases, including benign and malignant tumors, sharply increase the production of SAA, which contributes to its aggregation [28,29,30]. Under these conditions, the N-terminal fragment is cleaved from SAA and forms cytotoxic amyloid deposits in various organs and tissues [87].

The formation of amyloid fibrils of the calcitonin protein is also associated with tumors. Calcitonin is a hormone that decreases the renal tubular re-absorption of sodium, phosphate and calcium [89]. Its secretion or release by parafollicular cells of the thyroid gland is mediated by calcium and by gastrointestinal hormones, including gastrin and pancreozymin [90]. This hormone forms amyloid oligomers and aggregates in at least 80% of patients with medullary thyroid carcinomas. Moreover, this amyloid has also been found in patients with medullary carcinoma upon kidney biopsy, and in abdominal adipose tissue biopsy specimens [91]. Amyloidogenesis is mediated by increased levels of calcitonin production in medullary carcinoma. [37].

Odontogenic ameloblast-associated protein amyloidosis is related to Pindborg tumors and calcifying epithelial odontogenic tumors [92]. Normally, the odontogenic ameloblast-associated protein is implicated in diverse activities, such as ameloblast differentiation and enamel maturation [93]. This protein is up-regulated in calcifying epithelial odontogenic tumors, cervix cancer and gastric cancer [39]. Amyloid deposits are found in the maxilla or, more often, in the mandible in the region of an unerupted tooth [94].

It is likely that further research will identify new amyloids associated with cancers. In particular, there is some reason to believe that the protein Cathepsin K forms amyloid fibrils in angiomyolipoma [95]. However, these data have not yet received confirmation. Malignant transformation is accompanied by genomic instability and can provoke mutations that contribute to the amyloid formation by certain proteins. Several types of amyloidoses have been described that arise only against the background of mutations provoked by cancer, but this topic is beyond the scope of our review.

## 4. Infectious Agents, Chronic Pathologies, and Metabolic Diseases Cause Amyloidoses

Interestingly, the serum amyloid A protein forms pathological amyloid aggregates both during carcinogenesis and against the background of chronic infection and inflammation. Serum amyloid A (SAA) has a role in immune regulation, serves as an opsonin for bacterial phagocytosis and participates in the reverse cholesterol transport from injured tissues [87]. Chronic infections (such as tuberculosis, osteomyelitis and bronchiectasis), autoimmune diseases and chronic inflammatory disorders increase the production of SAA multifold [96]. Elimination of the infectious agent or the factor causing chronic pathology leads to a decrease in the level of the SAA protein production and a gradual loss of amyloid aggregates [97,98].

One of the factors provoking pathological amyloidoses is metabolic syndrome. This syndrome is characterized by the following metabolic disorders: abdominal obesity, high blood pressure, high blood sugar, high serum triglycerides, and low serum high density lipoprotein [99]. These pathologies provoke a change in the production of many proteins in various cell types [100]. Thus, it is not surprising that proteomic changes occurring against the background of this syndrome contribute to amyloidogenesis. AD and PD are much more common in patients with metabolic syndrome than in people of the same age in the control group [66,101]. Type 2 diabetes, closely associated with metabolic syndrome, also causes pathological amyloidogenesis. One of the indispensable characteristics of type 2 diabetes is the deposition of amyloid fibrils of islet amyloid polypeptide (IAPP) in the islets of Langerhans [102]. In this type of diabetes, the cells lose their sensitivity to insulin. The level of insulin secretion increases, which leads to a higher secretion of IAPP. These events promote IAPP oligomerization, fibril formation, and β-cell injury [31]. IAPP was shown to impair the blood–brain barrier and promote Aβ aggregation in a seeding-like manner [31,103]. This fact may explain why type 2 diabetes increases the risk of AD [104].

The formation of amyloid aggregates of apolipoprotein A-IV leads to the development of a rare systemic amyloidosis. This protein is produced by the enterocytes in the small intestine and has an important role in the absorption, transportation and metabolism of lipids [105]. This type of amyloidosis is detected mainly in elderly men [106]. The factors causing amyloidogenesis have not been characterized, but it is known that in newborn swine jejunum, a high-fat diet acutely induces a 7-fold increase in the apolipoprotein A-IV expression [36]. Thus, there is reason to believe that pathologies associated with impaired lipid metabolism can cause overproduction and amyloidogenesis of this protein.

The lung surfactant protein SP-C forms pathological amyloid aggregates against the background of pulmonary alveolar proteinosis (PAP) syndrome [38]. PAP is observed with lung infections (e.g., tuberculosis) and with exposure to a number of inhaled substances, including silica, aluminum dust, insecticides, and titanium [107,108]. This syndrome is characterized by the accumulation of alveolar surfactant and the dysfunction of alveolar macrophages [109]. Alveolar macrophages remove approximately half of the expelled surfactant by catabolism of phospholipids and efflux, and reverse transport of cholesterol to the liver [109]. In various diseases, the number of alveolar macrophages in the lungs decreases, which can contribute to the accumulation and aggregation of the SP-C protein.

Two more amyloidoses occur against the background of chronic kidney pathologies. Leukocyte chemotactic factor-2 amyloidosis (ALECT2) usually occurs in people who have chronic renal failure and mild urine sediment, with or without proteinuria [110]. Leukocyte chemotactic factor-2 plays a role in chemotaxis, cell proliferation, immunomodulation, damage and repair processes, and glucose metabolism [111]. Amyloid deposits of this protein most commonly affect the kidneys and liver [112]. This type of amyloidosis has been described recently, and the specific mechanism of amyloid fibrils formation in renal pathologies has not yet been studied. No doubt, it is associated with the induction of the stress response in chronic renal disease. β_2_-microglobulin amyloidosis also occurs in chronic kidney disease and is often seen in patients with long-term dialysis [113,114]. β_2_-microglobulin (β_2_m) is present on the surface of all nucleated cells and is involved in the presentation of peptide antigens to the immune system [115]. In patients with renal insufficiency, this protein accumulates in the serum and forms amyloid deposits, which leads to secondary osteoarticular destruction [113,114]. Multiple factors have been shown to enhance the aggregation of β_2_m in vitro and in vivo, including Cu2^+^, glycosaminoglycans, lysophosphatidic acid, non-esterified fatty acids, and collagen [35,116].

The presented data show that amyloidoses of wild-type proteins are a secondary event that is provoked by various pathologies.

## 5. Conclusions

Approximately 90% of all cases of amyloid diseases belong to the group of sporadic amyloidoses. Infectious and genetically determined amyloidoses are not so widespread, and their causes are obvious. In the first case, the disease develops as a result of the penetration of prion particles into the body, which causes the pathological conversion of the host’s own protein into amyloid conformation. In the case of genetically determined amyloidoses, the pathology is associated with the occurrence of mutations that provoke amyloidogenesis. The data presented in this review indicate that sporadic amyloidoses can be regarded as a secondary event occurring against the background of diseases provoking a cellular stress response (Table 1) For the majority of proteins included in this group, it was shown that certain factors cause their overproduction, which promotes amyloidogenesis (Table 1). In some cases, amyloidogenesis is a consequence of a local increase in the concentration of the protein or its modifications, which are also caused by stress (Table 1). The effects of pathological factors on the level of production, modification or local concentration of amyloidogenic proteins have not yet been characterized for only two sporadic amyloidoses (Table 1). For many proteins of this group, an increased frequency of the formation of pathological aggregates in elderly people is noted. At the same time, the analysis of specific factors provoking amyloidogenesis indicates that the trigger of these pathologies is certain diseases or physiological factors, but not aging itself. For example, the sporadic form of AD almost always occurs in people with vascular pathologies, head injuries, or metabolic diseases [45]. The only exception is the lactadherin protein, which is detected in amyloid form in blood plasma in 100% of people over the age of 50 [117]. The factors promoting the aggregation of lactadherin have not been characterized, and its amyloid conformers show no significant cytotoxicity. This amyloidosis should be regarded as a natural age-determined change rather than a pathology.

Taking into account the fact that various pathologies cause a stress response inducing amyloidogenesis, we can predict that the list of pathological amyloidoses will grow quickly. The amyloidoses known to date were mainly identified by chance. In recent years, proteomic screening methods have been developed that make it possible to identify amyloid-like proteins in any organs and tissues [118,119]. The use of these methods will enable the identification of pathological amyloids that arise against the background of various diseases. Important results can be obtained when searching for amyloids associated with various types of cancer. Each type of cancer provokes specific changes in the proteome, and it is expected that new cancer-associated amyloidoses will be described in the near future. Some amyloid proteins are already being considered as biomarkers of malignant tumors [120,121,122]. In addition, cancer-induced amyloidosis can cause toxic effects on healthy cells, and this factor should be taken into account in the study of various types of cancer.

The real extent of the spread of sporadic amyloidoses is likely much greater than it seems now. Detection of amyloidoses is generally performed with biopsy or by analyzing postmortem samples, but such approaches cannot fully reflect the real spread of amyloidoses. It is likely that the relationship between amyloidoses and many diseases has not yet been characterized. It is interesting to note that the systemic pathologies that occur after severe COVID-19 are similar to the clinical picture that is observed in systemic amyloidosis AA associated with aggregation of the SAA protein. In both cases, inflammatory processes occur in a wide variety of organs, such as the kidneys, liver, lungs, and heart, combined with disorders of the peripheral and central nervous systems [87,123]. It has been shown that an increase in the level of cytokines and in particular interleukin-6 is a biomarker of the severe course of COVID-19 [124]. Increased plasma levels of interleukin-6 and some other cytokines cause SAA overproduction [87]. Moreover, it has already been shown that the severe course of COVID-19 is accompanied by the overproduction of the SAA protein [125]. High serum levels of SAA are so far considered only as a biomarker for severe COVID-19. However, it is important to remember that SAA overproduction is the main factor causing AA amyloidosis [87]. Generalization of these data allows us to hypothesize that AA amyloidosis is a factor causing systemic disorders after severe COVID-19. Testing this hypothesis may be important since this amyloidosis is curable if diagnosed early.

Revision of modern ideas about the causes of pathological amyloids may also have prognostic value. Since the most common amyloidoses are triggered by vascular disease, metabolic disease, chronic inflammation, and cancer, progress in the treatment of these diseases will provide a major contribution to reducing the risk of amyloidosis. Elimination of the causes of pathological amyloids can facilitate their treatment. The development of drugs that block the growth and spread of amyloid fibrils, without eliminating the root causes of amyloidosis, is only auxiliary.

## Figures and Tables

**Table 1 ijms-22-04092-t001:** Risk factors for sporadic amyloidoses.

Sporadic Amyloidosis	Protein	Organs/ Tissue	Factors that Can Induce Amyloidogenesis	Effects of Pathological Factors on Amyloidogenic Protein Production/Modification/Local Concentration
Alzheimer’s disease ^1^	Aβ	CNS	Oxidative stress, vascular pathology, head injuries, metabolic syndrome, type 2 diabetes	Aβ overproduction under oxidative stress [20]
Parkinson’s disease ^2^	α-Synuclein	CNS	Head trauma, exposure to farming chemicals, oxidative stress	α-Synuclein overproduction in traumatic brain injury [21]
Tauopathies ^3^	Tau	CNS	Brain injury and chronic inflammation	Tau hyperphosphorylation in traumatic brain injury [22]
Non-inherited CJD and fatal insomnia ^4^	Prion Protein	CNS	Oxidative stress, hypoxia	Prion protein overproduction under oxidative stress and hypoxia [23,24,25,26]
Amyloidosis associated with preeclampsia ^3^	α-1 antitrypsin, albumin, Ig(L), ceruloplasmin, IFI6, Aβ	All organs except CNS	Disease of pregnant women associated with impaired angiogenesis of the uterus	N/D ^6^
AA amyloidosis ^3^	Serum amyloid A	All organs except CNS	Chronic infections, inflammations, different types of cancer	Serum amyloid A overproduction in chronic infections and cancer [27,28,29,30]
Islet amyloidosis ^3^	Islet amyloid polypeptide	Islets of Langerhans in pancreas	Type 2 diabetes	Islet amyloid polypeptide overproduction in type 2 diabetes [31]
Isolated atrial amyloidosis ^3^	Atrial natriuretic factor	Cardiac atria	Hypertension, myocardial infarction, cardiomyopathy, valve insufficiency	Atrial natriuretic factor overproduction in heart disease [32]
Senile systemic amyloidosis ^5^	Transthyretin	All organs except CNS	High level of glucocorticoids, psychosocial stress, oxidative stress	TTR overproduction in psychosocial stress and high glucocorticoid levels [33]
AH amyloidosis ^3^	Ig(H)	All organs except CNS	Multiple myeloma	Ig(H) overproduction in multiple myeloma [34]
AL amyloidosis ^3^	Ig(L)	All organs except CNS	Multiple myeloma and Waldenström’s macroglobulinemia	Ig(L) overproduction in multiple myeloma [34]
β_2_-Microglobulin amyloidosis ^5^	β_2_-Microglobulin	Musculoskeletal system	Chronic kidney disease and dialysis	β_2_-Microglobulin overproduction in renal insufficiency [35]
Apo A-IV amyloidosis ^3^	Apolipoprotein A-IV	Renal medulla and systemic	Dietary fat absorption	Apolipoprotein A-IV overproduction in high fat diet [36]
LECT2 amyloidosis ^3^	Leukocyte Chemotactic Factor-2	All organs except CNS	Chronic renal insufficiency	N/D ^6^
Calcitonin amyloidosis ^3^	Calcitonin	Thyroid gland, kidneys, fatty tissue	Carcinoma	Calcitonin overproduction in carcinoma [37]
Lung SP-C protein amyloidosis ^3^	Lung surfactant protein C	Lung	Lung infections, exposure to inhaled chemicals	Increased local concentration of the lung surfactant protein in lung infections [38]
OAAP amyloidosis ^3^	Odontogenic ameloblast-associated protein	Maxilla, mandible	Odontogenic tumors, cervix and gastric cancer	The odontogenic ameloblast-associated protein overproduction in odontogenic tumors, cervix cancer and gastric cancer [39]

^1^ Sporadic AD accounts for more than 95% of cases [40]. ^2^ Sporadic PD is 85–90% [41]. ^3^ For these proteins, only the sporadic form of amyloidosis has been characterized. ^4^ Sporadic prion diseases is approximately 85% [42]. ^5^ The ratio of cases of sporadic and other forms of amyloidosis for these proteins has not been characterized. ^6^ N/D—not determined.

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
