# Peer review of "Stress Response Is the Main Trigger of Sporadic Amyloidoses"

_ijms, 2021, doi:10.3390/ijms22084092_

Round 1

Reviewer 1 Report

The review by  Alexey P. Galkin et al. “Stress Response is the Main Trigger of Sporadic Amyloidoses” is focused on the role played by cellular stress condition in the onset of proteinopathies such  as Alzheimer’s disease, prion disease, type II diabete mellitus and so on. Authors focus the attention on the triggering role played by pre-existing pathological condition in the amyloidogenesis of protein involved in the most common misfolding diseases. Authors provided an interesting point of view in the field, reporting a good number of references. Review is well written and I suggest it for publication on International Jourmal of Molecular Science

Author Response

Thank you for your attention to our manuscript.

Reviewer 2 Report

In this review article, Galkin and Sysoev have proposed that stress response is the primary trigger of sporadic amyloidosis. Their arguments are based on recent studies linking onset of amyloidosis “after” the development of diverse pathological events, including chronic infection and metabolic impairment.

I have the following comments to share.

(a) In table 1, can the authors indicate the prevalence of the sporadic causes in each disease?

(b) For section 2, can the authors sub-divide the paragraphs according to the disease category; for example, Alzheimer’s, Parkinson’s prions?

(c) Line 105: Post-stroke survivors who subsequently developed dementia is likely to be suffering from vascular dementia. There are also group with mixed dementia. Perhaps the authors can provide greater clarification.

(d) Line 105: Are young individuals referred by the authors diagnosed with traumatic brain injuries (TBI)?

(e) Line 160-161: Is reference 26 a review article? Can the authors provide primary study that demonstrate PrP overexpression during hypoxic condition?

(f) Line 160: Similarly reference 25 is a review article. The authors can remove this reference since they have provided two primary studies (reference 23 and 24).

In conclusion, this review is addressing a broad topic involving many diverse diseases. Some were more extensively discussed than others. Nevertheless, the hypothesis is not new and have been addressed in other recent review articles.

Author Response

I have the following comments to share.

(a) In table 1, can the authors indicate the prevalence of the sporadic causes in each disease?

The answer: For most of the proteins presented in the Table 1, only the sporadic form of amyloidosis is described. Non-sporadic forms of amyloidosis were noted only for five proteins (Aβ, α-Synuclein, PrP, TTR and β2-Microglobulin). Data on the ratio of the sporadic and non-sporadic forms of these amyloidoses are now presented in the footnotes to the Table 1. Also, see the References, Lines 473-478.

(b) For section 2, can the authors sub-divide the paragraphs according to the disease category; for example, Alzheimer’s, Parkinson’s prions?

The answer: We have inserted subheadings in Section 2 as requested by the reviewer:

Line 93: 2.1. Sporadic Alzheimer's disease and tauopathies

Line 144: 2.2. Sporadic Parkinson's and prion diseases

Line 169: 2.3. Sporadic amyloidoses localized outside the CNS

(c) Line 105: Post-stroke survivors who subsequently developed dementia is likely to be suffering from vascular dementia. There are also group with mixed dementia. Perhaps the authors can provide greater clarification.

The answer: Unfortunately, the authors of the article only described cases when patients died soon after injury. In this regard, it is impossible to draw a conclusion about the development of dementia. (Section 2, Lines 108-109).

(d) Line 105: Are young individuals referred by the authors diagnosed with traumatic brain injuries (TBI)?

The answer: Yes. The young individuals were diagnosed with a traumatic brain injury. We have clarified this in the manuscript (Section 2, Lines 109-112).

(e) Line 160-161: Is reference 26 a review article? Can the authors provide primary study that demonstrate PrP overexpression during hypoxic condition?

The answer: In response to the reviewer's comment, we replaced the review article in the References with the article in which the original research was conducted. See the References, Lines 439-442.

(f) Line 160: Similarly reference 25 is a review article. The authors can remove this reference since they have provided two primary studies (reference 23 and 24).

The answer: We removed reference 25. See the References, Lines 436-438 and 443-444.

We have also provided more correct links when describing preeclampsia (Lines 199-200).

In conclusion, this review is addressing a broad topic involving many diverse diseases. Some were more extensively discussed than others. Nevertheless, the hypothesis is not new and have been addressed in other recent review articles.

The answer: The relationship of several sporadic amyloidosis with other diseases has indeed been addressed in some review articles. We refer to these works in the manuscript. The novelty of our work lies in the fact that we summarize for the first time all current data on the factors causing sporadic amyloidoses, and present the concept that ALL sporadic amyloidoses can be regarded as a secondary event occurring against the background of diseases provoking a cellular stress response.

English language and style are fine/minor spell check required

The answer: We've done a spell check and made minor fixes and clarifications:

Line 176: Abbreviation ANF instead ANP;

Line 75: We insert but in some cases instead but some cases;

Line 305: We insert kidneys instead kidney;

Table 1: We changed the word breaks from line to line in the table;

Table 1: All organs except CNS instead Kidney (in description of Senile Systemic Amyloidosis);

Table 1: Musculoskeletal system instead Kidney (in description of β2-Microglobulin amyloidosis);

Table 1: Renal instead Kidney (in description of Apo A-IV amyloidosis);

Table 1: All organs except CNS instead Kidney (in description of LECT2 amyloidosis);

Subsection 2.3, Lines 179-180: in the heart and other organs instead in the heart;

Section 3, Lines 211 and 234:  tumor instead carcinogenesis.

 Thank you for your attention to our manuscript.